# Non-Cationic RGD-Containing Protein Nanocarrier for Tumor-Targeted siRNA Delivery

**DOI:** 10.3390/pharmaceutics13122182

**Published:** 2021-12-17

**Authors:** Xiaolin Yu, Lu Xue, Jing Zhao, Shuhua Zhao, Daqing Wu, Hong Yan Liu

**Affiliations:** 1Georgia Cancer Center, Augusta University, Augusta, GA 30912, USA; xyu@augusta.edu (X.Y.); xue_lu@jlu.edu.cn (L.X.); dwu@cau.edu (D.W.); 2Department of Pediatrics Hematology, The First Hospital of Jilin University, Changchun 130021, China; 3Department of Gynecology and Obstetrics, The Second Hospital of Jilin University, Changchun 130041, China; zhaojing_2918@163.com (J.Z.); zhaoshuhua-1966@163.com (S.Z.); 4Center for Cancer Research and Therapeutic Development, Clark Atlanta University, Atlanta, GA 30314, USA; 5Dotquant LLC, Seattle, WA 98195, USA

**Keywords:** RGD, siRNA delivery, tumor targeting, tumor neo-vasculature, integrin α_v_β_3_, dsRNA binding domain

## Abstract

Despite the recent successes in siRNA therapeutics, targeted delivery beyond the liver remains the major hurdle for the widespread application of siRNA in vivo. Current cationic liposome or polymer-based delivery agents are restricted to the liver and suffer from off-target effects, poor clearance, low serum stability, and high toxicity. In this study, we genetically engineered a non-cationic non-viral tumor-targeted universal siRNA nanocarrier (MW 26 KDa). This protein nanocarrier consists of three function domains: a dsRNA binding domain (dsRBD) (from human protein kinase R) for any siRNA binding, 18-histidine for endosome escape, and two RGD peptides at the N- and C-termini for targeting tumor and tumor neovasculature. We showed that cloned dual-RGD-dsRBD-18his (dual-RGD) protein protects siRNA against RNases, induces effective siRNA endosomal escape, specifically targets integrin α_v_β_3_ expressing cells in vitro, and homes siRNA to tumors in vivo. The delivered siRNA leads to target gene knockdown in the cell lines and tumor xenografts with low toxicity. This multifunctional and biomimetic siRNA carrier is biodegradable, has low toxicity, is suitable for mass production by fermentation, and is serum stable, holding great potential to provide a widely applicable siRNA carrier for tumor-targeted siRNA delivery.

## 1. Introduction

siRNA has emerged as an invaluable tool for studying gene functions and developing treatments for intractable diseases, such as cancer [1,2,3], viral infection [4,5], and genetic disorders [6,7]. Generally, siRNA mediates posttranscriptional gene silencing by targeting and cleaving complementary mRNA [8,9,10]. The advantage of siRNA over small chemical drugs is that siRNA sequences can be rapidly designed for highly specific inhibition of the target protein expression [11]. siRNA also has advantages over antisense oligonucleotides. In one head-to-head comparison, siRNA knockdown of gene expression was about 100- to 1000-fold more efficient than antisense oligonucleotides (ODNs) [12]. In 2018, the FDA approved the first siRNA drug (patisiran), and in 2019, they approved the second siRNA drug (Givlaari) [13], which suggests that siRNA drugs are emerging as the third class of medicine after chemical drugs and antibodies for treating diseases by targeting the root cause. However, most siRNA drugs so far have focused on targeting the liver. Therefore, targeting delivery systems that can deliver siRNA to various target tissues other than the liver are needed.

To successfully deliver RNA, vectors should be designed to overcome barriers such as toxicity, off-target effects, endosomal entrapment, and ease of mass production. However, almost all current siRNA vectors (e.g., lipoplexes [4], polymers [14], inorganic nanoparticles [15], cell-penetrating peptides [16], and micelles [17]) are positively charged, and so are their siRNA complexes. Although cationic carriers facilitate their condensation of negatively charged siRNA and enable endosomal escape, cationic carriers can interact with negatively charged serum components, form unstable complexes in the circulatory system, and cause off-target effects [18,19]. Additionally, inorganic nanoparticles and polymers have serious safety concerns because they are non-degradable and have poor clearance. Therefore, current vectors are unsuitable for the systemic delivery of siRNA. In our previous studies, we developed an innovative technology for aptamer-siRNA delivery by engineering a non-cationic protein-based carrier (dsRBD-18 His) [20]. Our small protein carrier was developed by adding 18 histidine (His) peptide on a dsRNA binding domain (dsRBD) protein from human protein kinase R (PKR). DsRBD protein enables binding to dsRNA via specific 3D conformation but not via charge–charge interaction [21,22]. His molecules have a pKa value of about 6; at a neutral pH, they are uncharged, and they are charged in acidic conditions, such as in endosomes. We proved that 18His is capable of providing sufficient buffering capacity to drive cargo endosomal escape [20]. Other studies also showed that the addition of 2 His to protein enhanced endosomal escape [23], and two His mutations increased the engineered Her2-targeted antibody drug lysosomal delivery [3]. Notably, siRNA/dsRBD complexes are able to protect siRNA from degradation by ribonucleases [24]. Our developed dsRBD-18 His protein can bind siRNA and enables siRNA endosomal escape.

However, all current siRNA vectors, including dsRBD-18His, do not have cell-type specificity, and targeting molecules, such as aptamers, antibodies, or ligands, must be added onto siRNAs. That conjugation process is time-consuming and restricts siRNA application in vivo.

In this study, we developed a universal tumor-targeted siRNA vector by fusing RGD (arginine–glycine–aspartic acid) peptide to dsRBD-18His protein. RGD has high affinity and specificity for integrin α_v_β_3_, which is highly expressed on tumor neovasculature endothelial cells and many tumor cells but is not present in resting endothelial cells and normal organ systems. RGD peptide is a well-validated tumor-targeting molecule and has been used for guiding imaging agents [25,26,27] and drugs [28,29,30] for tumor diagnosis and therapy in clinical settings. Our cloned RGD- and 18His- containing protein vector with load-to-go capability will simplify siRNA delivery in vivo and enable delivering a wide range of siRNAs to many tumors and tumor microenvironments. To increase binding efficacy, we cloned two RGD domains flanking dsRBD-18His protein on both sides.

## 2. Materials and Methods

### 2.1. Materials

Cell culture products were purchased from Gibco through Thermo Fisher Scientific (Waltham, MA, USA). Cell lines were purchased from the American Type Culture Collection (ATCC, Manassas, VA, USA). PET28a plasmid, BugBuster mix, Ni-charged His Bind Resin, and *E. coli* BL21 (DE3) competent cells were ordered from Novagen/MilliporeSigma (Burlington, MA, USA). Restriction enzymes and Taq polymerases were obtained from New England Biolabs (Ipswich, MA, USA). Antibodies were ordered from Cell Signaling Technology (Danvers, MA, USA). siRNAs and fluorophore-labeled siRNAs were ordered from Dharmacon/Horizon Discovery (Lafayette, CO, USA). LysoTraker Green DND-26 was from Invitrogen (Carlsbad, CA, USA). Cell Counting Kit-8 reagent was ordered from Dojindo Molecular Technologies (Rockville, MD, USA).

### 2.2. Mouse

All animal studies were approved by the Institutional Animal Care and Use Committee at Augusta University. The approved animal protocol number was 2014-0658. Athymic nu/nu mice were purchased from Envigo Harlan Laboratories, Inc. (Indianapolis, IN, USA). The methods were carried out in accordance with the approved guidelines.

### 2.3. Construction of Fusion Protein Dual-RGD (Dual-RGD-dsRBD-18his) Expression Plasmid

The dsRBD-18His-containing plasmid was used as a PCR template. For amplification by PCR, the following primers were used: forward, 5′- AAA**GGATCC** ATG TGT GAT TGT CGT GGA GAT TGT TTC TGT GGTAGT GCTGGTGAT CTTTCAGCAG-3′; reverse, 5′-TT**CTCGAG**GCTGTCTCCACGGCCGTGGTGGTGGTGGTGGTGCACTGAGGTTTCTTCT-3′. *BamH1* and *Xho1* recognition sites were introduced in the primers and bolded. The PCR product and PET28a plasmid were digested with *BamH1* and *Xho1*. After digestion, the PCR fragment was inserted into PET28a by *T4 DNA* ligase. Ligates were transformed into *E. coli* BL21 (DE3) competent cells. The positive clones were confirmed by gene sequencing.

### 2.4. Expression and Purification of Dual-RGD Recombinant Protein

Positive colonies were selected and grown at 37 °C in LB medium containing kanamycin (30 µg/mL). When the OD_600_ reached 0.4–0.6, protein expression was induced by isopropyl-β-D-thiogalactopyranoside (IPTG, 1 mM) for 4 h at room temperature. Cells were harvested by centrifugation at 10,000× *g* for 10 min, and the pellet was stored at −20 °C. To isolate the soluble proteins, the cell pellet was lysed by the BugBuster mix in the presence of a protease inhibitor EDTA-free cocktail. The cell lysate was centrifuged at 20,000× *g* for 20 min at 4 °C to remove insoluble cell debris. The soluble extracts were loaded onto a nickel resin column and eluted with His•Bind^®^ buffer. The eluted protein was dialyzed against PBS containing 10% glycerol and 0.1% (*v*/*v*) β-ME for 24 h. Protein concentrations were determined with the Bio-Rad Protein Assay using BSA as the standard. Purified protein was probed using SDS-PAGE.

### 2.5. Measurement of Hydrodynamic Diameter and Charge

The hydrodynamic diameter and zeta potentials of purified monomer dual-RGD protein (0.1 mg/mL) in PBS buffer was detected by a Zetasizer Nano ZS analyzer (Malvern, Worcestershire, UK).

### 2.6. SiRNA Binding Assay

To assess siRNA binding capability of the dual-RGD carrier, a gel retardation assay was performed. EGFR siRNA was labeled with FAM fluorophore at the 5′ end of the sense strand. FAM-siRNA was incubated with dual-RGD at different protein/siRNA molar ratios (1:1, 2:1, a d 4:1) for 30 min at room temperature. Bound and unbound siRNA was quantified using 1% agarose gel electrophoresis. Images first were captured under a UV transilluminator. Then, the same gel was stained with Coomassie blue to reveal the protein location, and the stained gel was imaged again under bright light.

### 2.7. RNase Resistance and Serum Stability

siRNA was labeled with FAM at the 5′ terminus of the sense strand. Dual-RGD/siRNA complex or siRNA only was spiked with 50% (*v*/*v*) fresh pooled human serum at different times (1 h, 4 h, and 24 h); 1% agarose gel electrophoresis was performed to monitor the dissociation.

### 2.8. Western Blot

Cell lysates or tumor tissues were lysed in lysis buffer (M-PER Mammalian Protein Extraction Reagent, Thermo Fisher Scientific) containing 1× Halt Protease Inhibitor Cocktails. The cell lysates were kept on ice for 40 min, vortexed 3 times, and centrifuged at 12,000× *g* for 10 min at 4 °C. The concentration of supernatant protein was determined with the Bio-Rad Protein Assay. Samples were separated on 8% SDS-PAGE and transferred to the PVDF membrane. The membranes were blocked with 5% non-fat milk and then incubated with primary antibodies for 2 h at room temperature, followed by incubation with quantum dot-conjugated secondary antibodies for 2 h at room temperature. The blots were imaged under a UV table. Western blot was quantified using ImageJ version 10.2 (NIH).

### 2.9. Cell-Type Specific Binding Assay

Cell lines with different integrin α_v_β_3_ expression levels were grown to 80% confluence and trypsinized from the dishes. The collected cells were fixed in 10% buffered formalin for 30 min. The siRNA sense strand was labeled with fluorophore FAM or Cy5 at the 5′ end. FAM-siRNA (5 nmoles of siRNA) was loaded into dual-RGD at a molar ratio of 1:1. The complex of dual-RGD/FAM-siRNA was incubated with fixed cell lines at 37 °C for 1 h, and the binding specificity of dual-RGD was detected by flow cytometry (BD FACSCalibur Cell Analyzer). In the competitive assay, the RGD peptide (10 nmoles) was treated cells for 1 h before adding the complex of dual-RGD/Cy5-siRNA.

### 2.10. Cellular Uptake and Endosomal Escape by Confocal Microscopy

MDA-MB-231 (integrin α_v_β_3_-positive) and C4-2 cells (integrin α_v_β_3_-negative) were seeded on 35 mm glass-bottom Petri dishes (MatTek Corp, Ashland, MA, USA) at a density of 2 × 10^4^ cells/well for 24 h. Complexes of Cy5-siRNA/dual-RGD or Cy5-siRNA/dsRBD-18His at 0.5 µM were added to the culture media for 6 h. LysoTracker Green DND-26 (80 nM) and DAPI (0.2 µg/mL) were added into the culture media for 2 h. Images were captured on a confocal laser scanning microscope (LSM 510, Carl Zeiss, Germany).

### 2.11. Tumor-Targeting and Biodistribution

Five-week-old female athymic nu/nu mice were injected with MDA-MB-231 cells (5 × 10^6^) mixed with Matrigel (*v*/*v* 1:1) (Corning, NY, USA) subcutaneously in their backs. After 4 weeks, tumor-bearing mice were tail-vein injected with 100 µL of Cy5-EGFR siRNA/dual-RGD complex (5 nmoles) or equal moles of Cy5-siRNA/dsRBD-18His complex. At time points of 3 h, 6 h, 12 h, and 24 h, the Cy5 fluorescence signals of mice were detected with the Xenogen IVIS100 imaging system (Caliper Life Sciences, Waltham, MA, USA) to monitor the siRNA distribution in the whole body. At 12 h and 24 h, major organs were removed and homogenized in buffer (10 mM Tris pH 7.4 and 0.5% Triton X-100) with a mortar and pestle at a ratio of 100 mg of tissue per mL of buffer. Subsequently, 100 μL of tissue homogenate was loaded into a 96-well plate. The plate was measured by a Tecan Infinite F200 Pro Microplate Reader (Tecan Company, Mannedorf, Switzerland).

### 2.12. In Vivo Gene Knockdown

MDA-MB-231 tumor-bearing mice (*n* = 5 per group) were iv treated with saline, EGFR siRNA only, EGFR siRNA/dsRBD-18His, or EGFR siRNA/dual-RGD twice a week for four weeks. Five nanomoles of siRNA were added in each group. The molar ratio of siRNA to dual-RGD or dsRBD-18His was 1:1.

### 2.13. Cell Viability

The proliferation and cytotoxicity of dual-RGD protein were quantified by measuring WST-8 formazan using Cell Counting Kit-8 (CCK-8). C4-2 and HEK293 cells in 10% FBS containing RPMI 1640 culture media were seeded in 96-well plates at a density of 2 × 10^3^ in a 5% CO_2_ incubator for 24 h at 37 °C. Cells were treated with varying concentrations of dual-RGD for 72 h. CCK-8 reagent (10 µL/well) was added to each well for 4 h. Absorbance was measured at 450 nm on a Tecan infinite M200 microplate reader (Tecan Company, Mannedorf, Switzerland).

### 2.14. In Vivo Toxicity Assay

Athymic mice (*n* = 6 per group, 3 male and 3 female) were iv treated with saline or dual-RGD (10 nmoles) or an equal volume of saline twice a week for 4 weeks. Major organs were collected for histology assay.

### 2.15. Histology Assay

Tumors and organs (spleen, lung, kidney, intestine, heart, liver, and brain) were collected and fixed with 4% paraformaldehyde. Sections (6 mm) were cut and mounted on slides and deparaffinized in xylene and ethyl alcohol. Major organs were stained with hematoxylin for 10 s and eosin for 30 s followed by dehydration. For tumor IHC assay, sections were blocked with 3% goat serum for 2 h and incubated with anti-EGFR antibody (1:500). After washing, the sections were incubated with biotinylated secondary antibody (1:200) (Vector Labs, Burlingame, CA, USA) for 1 h. Following washing, the sections were incubated with VECTASTAIN ABC reagents (Vector laboratories, Burlingame, CA, USA) for 30 min. The images were captured with a Nuance fluorescence microscope with a bright field imaging system (Caliper LifeSciences, Waltham, MA, USA). IHC quantification was performed with the ImageJ plugin IHC profiler (NIH) version 10.2. Images were changed to an 8-bit grayscale type and inverted under the “Edit” menu of ImageJ. After inverting, the DAB stain areas were bright, and the unstained areas were dark. The mean intensity was measured. 8–10 fields of each treatment group were assessed.

### 2.16. Statistical Analysis

The results were expressed as means ± SEM. All data were analyzed using a two-tailed Student’s *t*-test (Graph Pad Prism version 9) by comparing with the control group, and *p* < 0.05 was considered statistically significant.

## 3. Results

### 3.1. Construction and Expression of Dual RGD-dsRBD-18his (Dual-RGD) Recombinant Protein

In our previous studies, we cloned a dsRBD-18 His protein that contained a double-stranded RNA binding domain and 18His [20]. In this study, we genetically engineered two RGD peptides into N- and C- termini of dsRBD-18His protein, respectively. The N-terminal of dsRBD-18His was appended a cyclic RGD peptide (CDCRGDCFC (RGD-4C)). RGD-4C was cloned with TNFα and tumor necrosis factor-related apoptosis-inducing ligand (TRAIL). RGD-4C-containing TNFα and TRAIL have shown strong tumor selectivity and binding affinity, and also retain natural activities of TNFα [31] or TRAIL [28]. Meanwhile, a G**RGD**S (Gly-Arg-Gly-Asp-Ser) peptide was added to the C-terminal of dsRBD-18His since G**RGD**S is relatively conserved in natural RGD-containing proteins, such as fibronectin and α-lytic protease [32,33]. The gene structure of dual-RGD is illustrated in Figure 1A. Briefly, the dsRBD-18His gene in PET28a plasmid [20] was selected as the cloning PCR template. RGD genes and *BamH1*/*Xhol1* restriction enzyme recognition sites were introduced at the 5′ and 3′ ends of the primers, respectively. The recombinant protein was expressed in *E. coli* BL21 (DE3), induced by 1 mM of IPTG. Expressed protein was purified by an Ni-NTA affinity column. The purified protein was analyzed on SDS-PAGE with Coomassie blue staining. As shown in Figure 1B, there are two high-density bands in the non-reducing condition (Lane 2). After adding β-mercaptoethanol (β-ME) and heating, the larger-sized protein disappeared, and the small-sized protein remained (Lane 3), which indicates that dual-RGD protein can form a homodimer under nonreducing conditions. SDS-PAGE with a molecular weight (MW) marker showed the expected molecular weights of ~26 kDa (monomer) and ~52 kDa (dimer). On the basis of the above protein conformation, in the following protein purification, we added β-ME to keep dual-RGD protein as a monomer for function characterization. Furthermore, the hydrodiameter and charge were measured by a ZetaSizer analyzer. As shown in Appendix A, the hydrodynamic diameter of the monomer dual-RGD protein is about 23.5 nm, and the zeta potential is −7.85 mv. This indicates that the new dual-RGD protein is nano-sized with a slightly negative charge that is close neutral. This feature is in contrast with cationic polymer carriers.

### 3.2. siRNA Binding Capability

Next, we evaluated the functionalities of three domains in dual-RGD protein, including dsRBD, 18His, and RGD. First, we detected the dsRBD domain for siRNA binding. Selected as a siRNA model, EGFR siRNA was labeled with FAM fluorophore at the 5′ end of the sense strand. FAM-labeled EGFR siRNA was incubated with dual-RGD at different protein/siRNA molar ratios (1:1, 2:1, and 4:1) for 30 min at room temperature. Bound and unbound siRNA was quantified using 1% agarose gel electrophoresis. As shown in Figure 2A (left), under UV transilluminator, the mobility of the complex of siRNA and dual-RGD was much slower than the control of siRNA only. At a 1:1 molar ratio, all siRNA was bound to dual-RGD, and no free siRNA was detectable. To further detect if slowly moving siRNA had indeed bound to dual-RGD protein during the migration, the same gel was stained with Coomassie blue to identify the location of dual-RGD protein. As shown in Figure 2A (right), the bright field image shows that the protein location (Coomassie blue) was colocalized with the siRNA location (UV-revealed). The result clearly indicates that the dsRBD domain in dual-RGD maintains siRNA binding capability. In the previous studies [34] with cognate HIV TAR RNA (59NT), which contains bulges and an internal loop, using isothermal titration calorimetry, a model was fit for the two binding sites in TAR for the interaction of TAR RNA with dsRBD. One is a single high-affinity site and the second is a low-affinity site. At a 1:1 molar ratio of dsRNA to dsRBD, a type 1 complex with fast gel mobility was formed. At a higher ratio of 1:2, a type 2 complex was formed with slow gel mobility. The dissociation constant for the type 1 complex was 70 nM with a 1:1 stoichiometry. By contrast, the dissociation constant for the type 2 complex was 23 µM with 1:2 stoichiometry, which is 300-fold higher than that of the type 1 complex. Importantly, the previous studies [22] also demonstrated that: 16–20 bp dsRNA binding to dsRBD resulted in only the formation of complex 1 at a 1:1 molar ratio, while 22–24 bp dsRNA binding to dsRBD could form complex 2 at a molar ratio 1:2 of dsRNA to dsRBD. In our report, we only observed complex 1 (1:1 molar ratio) (Figure 2A) because our siRNAs were 19 bp [35]. Our results are in agreement with those of previous studies. Meanwhile, complex 1 had a smaller size than complex 2, making it suitable for tissue penetration in vivo.

### 3.3. Protection of siRNA against Nucleolytic Degradation

Following the confirmation of dual-RGD/siRNA binding, we evaluated whether the formation of complexes could protect siRNAs from RNase attack. Dual-RGD/siRNA complex or siRNA only was spiked with 50% fresh pooled human serum at different times. As shown in Figure 2B, siRNA in a complex with dual-RGD did not show degradation for the 24 h test period, while siRNA alone in 50% human serum showed degradation at the start (1 h). Therefore, the dual-RGD carrier could protect siRNA against nuclease digestion. Notably, we did not perform chemical modification for the test EGFR siRNA. The serum volume always accounts for 50% (*v*/*v*) in all test groups. Thus, it is not surprising that the serum volume in siRNA-only groups was lower than that in siRNA/dual-RGD complex groups since siRNA alone has a lower volume than the siRNA/dual-RGD complex (Figure 2B).

### 3.4. RGD Binding Specificity

The binding specificity of dual-RGD protein was assessed in a series of cell lines with different levels of integrin α_v_β_3._ To verify the expression levels of integrin α_v_β_3_, different cell lines were probed for integrin β_3_ expression with Western blot. As shown in Figure 3A, the cell lines MDA-MB-231, Hs587T, and BT20 expressed high levels of integrin β_3_, while the cell lines C4-2, BXPC3, and HEK293T showed very low expression levels of integrin β_3_. Next, FAM-EGFR siRNA was loaded into dual-RGD at a molar ratio of 1:1. A complex of dual-RGD/siRNA was incubated with formalin-fixed cell lines at 37 °C for 1 h, and the binding specificity of dual-RGD was detected by flow cytometry. Notably, fixed cells allow RGD binding to target cells but do not induce endocytosis, thus providing a precise assessment of cell surface integrin receptor-RGD binding. As shown in Figure 3B, dual-RGD had low or no binding to C4-2 and HEK-293T cells, while it had a high binding intensity for MDA-MD-231, Hs578T, and BT-20. The correlation of single-cell fluorescence intensity and integrin β_3_ protein levels demonstrated the binding specificity of the RGD domain in dual-RGD protein. Furthermore, in another independent experiment, a competitive binding assay was performed to evaluate RGD receptor-specific binding. MDA-MB-231 cells were incubated with dual-RGD/siRNA in the presence of free RGD peptide. As shown in Figure 3C, after blocking with RGD peptide, the binding intensity of dual-RGD/siRNA to MDA-MB-231 cells was significantly reduced, which is displayed by the fluorescence peak’s significant shift to the left. The above experiments suggest that the binding of the dual-RGD/siRNA complex to the cells occurs through the fused RGD domain in the dual-RGD carrier.

### 3.5. Cellular Uptake and Subcellular Distribution of Dual-RGD Carrier

Furthermore, we assessed whether RGD-integrin binding induced cell-type-specific internalization and endosomal escape, which are the prerequisites for siRNA to trigger gene silencing. In this recombinant protein, dual-RGD was incorporated with 18His, which was expected to provide endosomal rupture through the proton sponge effect. To prove 18His endosomolytic activity, confocal microscopy was performed to visualize the subcellular location of siRNA. First, Cy5 (red) fluorophore-labeled siRNA was complexed with dual-RGD or dsRBD-18His (without the RGD-targeting moiety as a control) at a molar ratio of 1:1. Then integrin α_v_β_3_-positive MDA-MB-231 live cells were treated with siRNA/dual-RGD or siRNA/dsRBD-18His for 6 h in 37 °C in a CO_2_ incubator followed by a DPBS wash. In the meantime, nuclei were labeled with DAPI (blue), and endosomes/lysosomes were stained with LysoTracker (green). As shown in Figure 4 (top two layers), siRNA/dual-RGD-treated MDA-MB-231 cells showed a much higher density of the red fluorescence signal in the cytoplasm compared with the siRNA/dsRBD-18His-treated control. It is worth mentioning that the yellow fluorescence signal is derived from the colocalization of Cy5-siRNA (red) with LysoTracker (green) and indicates siRNA endosomal entrapment. In siRNA/dual-RGD-treated MDA-MB-231 cells, a low amount yellow fluorescence signal indicates that less Cy5-siRNA was entrapped in endosomes, and a high amount of red signal in the cytoplasm around the nuclei indicates that most siRNA has escaped from endosomes. The above results suggested that cloned RGD peptide can lead to receptor-mediated endocytosis, and internalized siRNA can escape from endosomes and diffuse to the cytoplasm. Furthermore, as a cell type control, integrin α_v_β_3_-negative C4-2 cells were treated with Cy5-siRNA/dual-RGD for 6 h and washed with DPBS. The confocal image (Figure 4, bottom layer) showed that there was no detectable Cy5-siRNA signal in C4-2 cells, which is consistent with the binding specificity assay shown in Figure 3. These results strongly imply that dual-RGD-directed siRNA endocytosis is cell type-specific, which may avoid off-target effects during in vivo circulation.

### 3.6. Target Gene Knockdown In Vitro

After confirmation of dual-RGD led siRNA endocytosis and endosomal escape, we evaluated whether siRNA triggered target gene knockdown. As a model, EGFR siRNA was selected and formed complexes with dual-RGD at a molar ratio of 1:1. EGFR siRNA/dual-RGD complexes were incubated with MDA-MB-231 cells or Hs587 T cells for 72 h. It is worth mentioning that both MDA-MB-231 and Hs587T cells have high EGFR expression, as reported in our previous studies [36]. Cationic lipofectamine RNAi MAX was used as a control to verify siRNA functionality. The gene knockdown was probed in cell lysates with Western blot. As shown in Figure 5, the dual-RGD/siRNA complex displayed a dose-dependent gene knockdown in both cell lines. At 200 nM, siRNA/dual-RGD could silence 75% of EGFR in MDA-MB-231 cells and 65% of EGFR in Hs578T cells. It is not surprising that lipofectamine RNAi MAX showed better performance. However, siRNA uptake by cationic lipofectamine occurred via charge–charge interaction between lipofectamine (cationic) and charged cell membranes (anionic). Since cellular uptake through charge–charge interaction has no cell-type specificity, lipofectamine is incapable of in vivo siRNA targeting delivery.

### 3.7. Tumor-Targeting Capability In Vivo

After confirming the functionalities of the three domains of dual-RGD protein in vitro, we further evaluated them in vivo. MDA-MB-231 tumor-bearing mice were intravenously (iv) injected with 100 µL of Cy5-EGFR siRNA/dual-RGD complex (5 nmoles) or equal moles of Cy5-siRNA/dsRBD-18His complex as a control. At time points of 3 h, 6 h, 12 h, and 24 h, Cy5 fluorescence signals in mice were detected with the Xenogen IVIS 100 imaging system to monitor siRNA distribution in the whole body (Figure 6). After the 3 h injection, the Cy5 siRNA signal could be detected in the tumors of dual-RGD-treated mice but not in the tumors of dsRBD-18His-treated mice. At the 6h time point, Cy5-siRNA in dual-RGD-treated mice could be clearly visualized in the tumors but with reduced signal around other organs, whereas Cy5-siRNA in dsRBD-18His-treated mice remained undetectable in tumors. The Cy5-siRNA signal in tumors could last up to 12 h in dual-RGD-treated mice. After 24 h, both dual-RGD- and dsRBD-18His-delivered Cy5-siRNA had been removed from the mice. These results indicate that dual-RGD enables siRNA to be delivered to tumor sites, where siRNA can remain for 12 h. Since we labeled the siRNA sense strand with Cy5, it is possible that the sense strand (passenger strand) of siRNA was degraded after it was loaded into the RISC (RNA-Induced Silencing Complex) followed by gene silencing, which is a natural process for RNA interference but not serum nuclease-induced degradation. The time-course imaging demonstrated that dual-RGD can guide siRNA, targeting tumors in vivo. At 12 h and 24 h, we removed the major organs from treated mice and homogenized the tissues. The homogenized tissues were detected by a fluorescence microplate reader, as shown in Appendix A; at 12 h, there were weak fluorescence signals in the sites of the intestine and the brain (around the eyes), and most organs, including the heart, lungs, kidneys, spleen, and liver, were free of Cy5 signals. We reason that the capillary blood vessels around the eyes have a slower metabolic rate than other organs may experience the process of removing all metabolites; thus, some signals were detected in these organs. At 24 h, almost all fluorescence signals had been cleared from control and dual-RGD mice.

### 3.8. Target Gene Knockdown In Vivo

Furthermore, target gene knockdown was assessed in MDA-MB-231 xenografts. When tumors reached 100 mm^3^, mice were iv treated with EGFR siRNA/dual-RGD (5 nmoles) or controls of saline, EGFR siRNA, or EGFR siRNA/dsRBD twice a week for 4 weeks. At the endpoint, the tumors and major organs were removed. Tumors were analyzed by IHC and Western blot. As shown in Figure 7. IHC demonstrated a significantly reduced expression in the siRNA/dual-RGD group, but not in control groups of saline, siRNA only, or siRNA/dsRGD-18His. The Western blot assay with tumor cell lysate proved the down-regulation of EGFR protein. ImageJ quantitation indicated that after the 4-week treatment, only 30% of EGFR protein remained. These results indicate that the target gene was significantly knocked down in vivo. As a comparison, EGFR expression in the brain was also detected by Western blot. As shown in Appendix A, there was no significant difference in EGFR expression between saline-treated and siRNA/dual RGD-treated mouse brains. This result indicates that dual-RGD did not induce gene knockdown in non-target organs.

### 3.9. Toxicity Assessment

Next, we evaluated the possible cytotoxicity of dual-RGD against integrin α_v_β_3_-negative cells. Since RGD as a drug has shown cytotoxicity for integrin α_v_β_3_-expressing tumor cells [37,38], we aimed to find whether dual-RGD protein has cytotoxicity against integrin α_v_β_3_-negative cells. Integrin α_v_β_3_-negative C4-2 and HEK293T cells were selected and treated with the varying concentrations of dual-RGD carrier for 72 h; then, CCK-8 reagent was added to detect cell viability. As shown in Figure 8, both C4-2 and HEK293T cells kept over 92% cell viability after being treated with 0.2–1.0 µM of dual-RGD carrier protein. The results showed that there was no significant toxicity in the detection range up to 1.0 µM, which is three times as high as the one used in the delivery work.

Furthermore, we evaluated the possible toxicity in vivo. After mice were iv treated with dual-RGD protein (10 nmoles) or saline twice a week for 4 weeks, major organs, including the brain, heart, kidneys, liver, lungs, intestine, muscle, and spleen, were collected. And the histopathology changes were examined. As shown in Figure 8B H&E staining revealed that there was no discernible abnormality observed in dual-RGD protein-treated groups compared with the saline-treated control. These results suggest that the human-origin dual-RGD protein carrier is biocompatible and shows no or low systemic toxicity in vivo.

## 4. Discussion

Tumor-targeted siRNA delivery is of great interest in medical research. It is highly desirable but technically challenging to engineer efficient vectors that are non-cationic, non-toxic, endosomal escapable, cell-type specific, and easy to mass-produce. To find a carrier for in vivo systemic siRNA delivery, natural human proteins are attractive because they are biocompatible, biodegradable, easy to clear from the body, and easy to mass-produce by fermentation.

Human PKR has been extensively studied [24,39] as a natural carrier for in vivo siRNA delivery. Natural PKR in the body is activated by dsRNA and plays a major role in the cellular antiviral response through the inhibition of eukaryotic initiation factor 2 (eIF-2) [40,41]. PKR consists of an N-terminal dsRBD, including a pair of double-stranded RNA binding motifs (dsRBM1 and dsRBM2) [42,43] and a C-terminal kinase domain [44]. dsRBM1 has a dominant role in molecular recognition of short dsRNA sequences (15–30 bp), whereas both motifs contribute to binding longer dsRNA sequences (>40 bp). dsRBD binds RNA through protein 3D conformation but not through charge–charge interaction [21]. In addition to its ability to sense dsRNA, primarily of viral origin, PKR is also activated in response to endogenous RNA, such as microRNAs [45,46]. dsRBD from PKR can bind to a broad range of dsRNA in a sequence-independent manner [47], which is unique among known RNP (ribonucleoprotein) complexes. It also has been proven that dsRBD only binds to dsRNA but not RNA-DNA hybrids or dsDNA [22], which provides the basis to generate dsRBD as an siRNA carrier. However, dsRBD lacks endosomal escape capability. To address this problem, we used polyhistidine as an endosomal escape moiety instead of peptide transduction domains (PTD) because PTD is also an arginine- and lysine-rich positively charged peptide. Histidine has a pKa value of about 6.0; at neutral or tumor environment (pH 6.5–6.9) conditions [48], they are mainly deprotonated (uncharged), while in acidic conditions, such as in late endosomes (pH 4.5–5.5) [49], histidine becomes protonated (charged) and facilitates osmotic swelling, leading to cargo release, a mechanism proposed as the proton sponge effect [50]. However, it is well known that the 6xHis tag used for protein purification does not demonstrate the capability of endosome escape. We reason that 6xHis is too short to offer enough buffering effect, and certain elongated His tags should confer cargo endosomal escape. We inserted 18 His peptide into dsRBD protein [20]. We compared the endosomal escape capability of dsRBD-18His and dsRBD-6His. Our results demonstrated that adding 12 more His could significantly improve endosomal escape capability. 18His can confer adequate buffering capacity to drive cargo endosomal escape. In contrast to cationic carriers, dsRBD-18His is uncharged in the physiological condition, has low toxicity, and is biodegradable. Based on the dsRBD-18His vector, we further equipped it with two RGD peptides for tumor targeting.

The mechanism of siRNA released from dsRBD is believed to occur via displacement by the highly abundant longer RNA molecules in the cytosol [51]. Since dsRBD recognizes RNA in a sequence-independent fashion, in the referred studies, with total RNA isolated from cells, dsRBD-siRNA complex was incubated with 80 ng/µL cellular RNAs (which is still much lower than the cytosolic RNA concentration), and the siRNA-dsRBD complex is totally disrupted. These studies support that the competitive binding and exchange of cellular RNAs to dsRBD can trigger siRNA release from dsRBD inside of cells.

In this study, we aimed to establish a general tumor-targeted siRNA delivery platform. We focused on the characterization of dual-RGD performance in vivo and in vitro, and we will evaluate tumor treatment efficacy in the next study. RGD peptides have been widely used in drug delivery and imaging and have minimized safety concerns [26,52]. RGD has high affinity and specificity for integrin α_v_β_3_, which is highly expressed in tumor new blood vessels and many tumor cells but is not present in resting endothelial cells or normal organ systems. New RGD carriers will simplify the process of siRNA delivery and enable targeted delivery of siRNAs to tumor blood vessels and many tumors, such as glioblastomas and melanomas and pancreatic, ovarian, breast, and prostate cancers [53,54,55]. RGD-based materials have undergone rapid development for sensitive tumor detection and tumor-targeted drug delivery in vivo [23,56,57]. RGD-PET (positron emission tomography) has been used in the clinic for tumor diagnosis [58,59]. RGD has also been widely used in developing anti-angiogenesis therapeutics [60,61] and conjugation with radionuclides [62,63] or chemotherapeutic drugs [64,65] for tumor treatment. TNFα has been fused with RGD by recombinant DNA technology and has achieved tumor targeting and reduced system damage [66,67]. Cyclic RGD peptide has shown better targeting capability in vivo. In this study, to avoid unwanted internal disulfide bonds between the N-terminal and C-terminal, we cloned one end (N-) with cyclic RGD and another end (C-) with linear GRGDS peptide. The results proved that the cloned protein possesses ideal functionality.

In terms of delivery efficacy, lipofectamine outperformed the dual-RGD carrier. This was expected since cationic charge–charge interaction between lipofectamine and the cell membrane is much more effective in gene transfection than ligand–receptor-mediated endocytosis, which is the assumed mechanism of the dual-RGD carrier. However, cationic lipofectamine cannot be used for in vivo cell-type-specific siRNA delivery because of nonspecific binding.

Cyclic peptide c(-RGDfV-), as the drug Cilengitide, has shown the inhibition of tumor growth and angiogenesis [37,38]. We envision that the fusion protein dual-RGD carrier complexed with siRNA will have increased anti-tumor efficacy in addition to siRNA-induced gene silencing.

## 5. Conclusions

Our developed protein-based dual-RGD vector has three functions: a dsRBD domain for siRNA docking, 18 His for endosomal escape, and RGD for tumor targeting. This three-in-one multidomain vector will address the problems of current siRNA carriers of positive charge, low serum stability, poor clearance, cytotoxicity, and lack of cell-type specificity. RGD is much better than transferrin and folate, which can only target cancer cells but cannot target the tumor neo-vasculature. Our cloned dual-RGD protein carrier with load-to-go capability will expedite siRNA translation to in vivo tumor therapy.

## Figures and Tables

**Figure 1 pharmaceutics-13-02182-f001:**
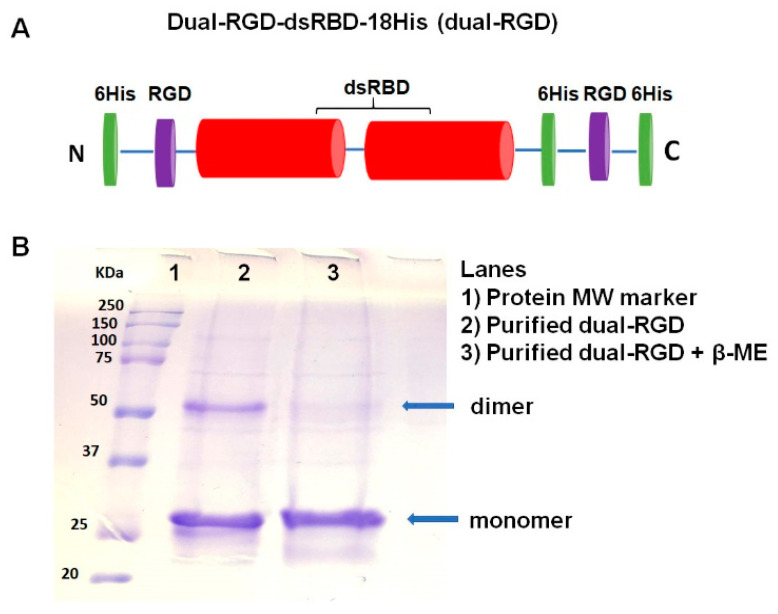
Cloning and production of dual-RGD-dsRBD-18H (dual-RGD) protein. (**A**) Schematic illustration of dual-RGD protein structure. (**B**) SDS-PAGE analysis of dual-RGD protein. Dual-RGD protein is shown as a monomer (26 KDa) and a dimer (52 KDa) in nonreducing conditions (lane 2) and a single monomer under reducing (lane 3) conditions.

**Figure 2 pharmaceutics-13-02182-f002:**
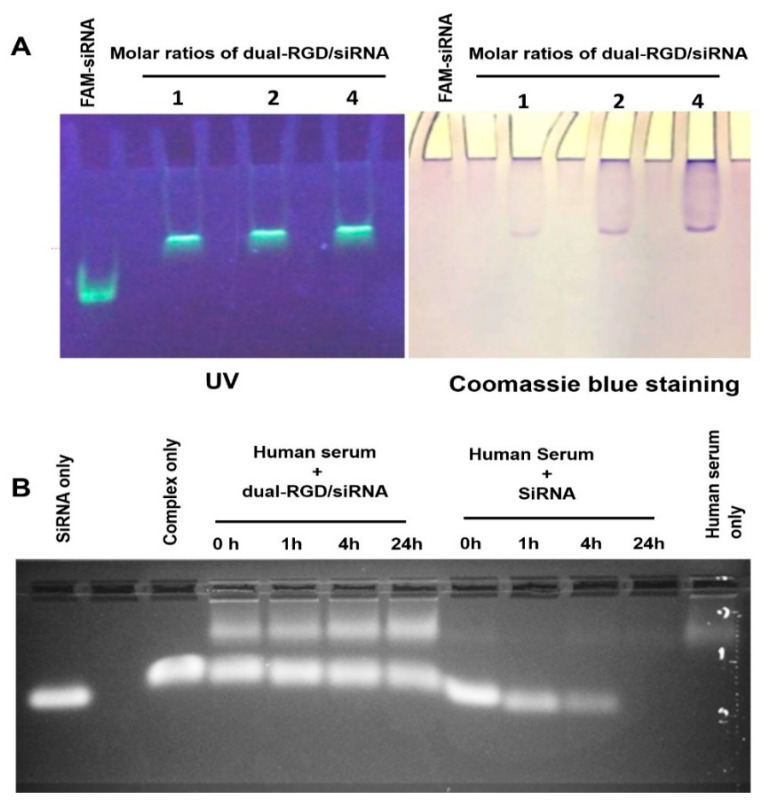
Characterization of dual-RGD on binding and protection of siRNA. (**A**) Gel retardation to assess dual-RGD/siRNA binding. FAM-labeled siRNA was incubated with different molar ratios of dual-RGD protein. The complexes were detected by 1% agarose electrophoresis. SiRNA migration location was imaged under a UV transilluminator. To localize dual-RGD protein position, the same gels were stained with Coomassie blue. Two images show that dual-RGD and siRNA were at the same location, indicating dual-RGD binding to siRNA. (**B**) Protection of siRNA against nucleases. Naked siRNA or siRNA/dual-RGD complexes were incubated with 50% (*v*/*v*) human serum for different time periods at 37 °C. The siRNA degradation was probed using agarose gel electrophoresis.

**Figure 3 pharmaceutics-13-02182-f003:**
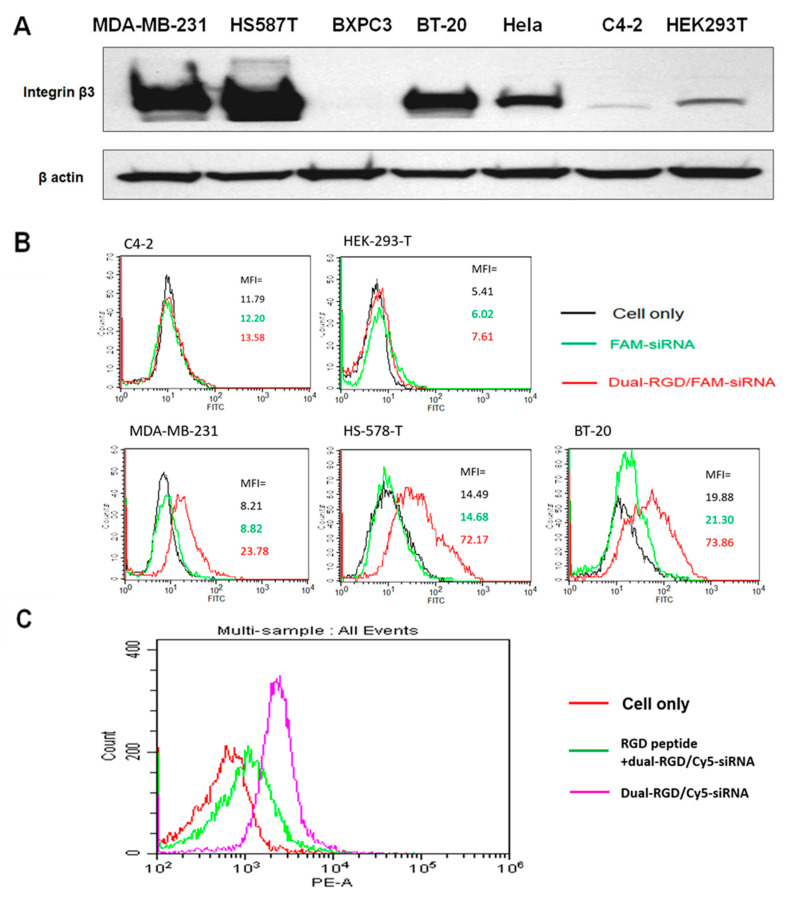
Integrin α_v_β_3_ binding specificity. (**A**) Detection of integrin **β_3_** expression levels in cell lines by Western blot. (**B**) Evaluation of dual-RGD cell type-specific binding by flow cytometry. FAM-siRNA/dual-RGD or FAM-siRNA only was incubated with formalin-fixed cell lines at 37 °C for 1 h, and the binding specificity of dual-RGD was detected by flow cytometry. (**C**) Competitive assay. Fixed cell lines were first treated with free RGD peptide, and then Cy5-siRNA/dual-RGD complexes were added to the cells. The binding intensity was detected by flow cytometry.

**Figure 4 pharmaceutics-13-02182-f004:**
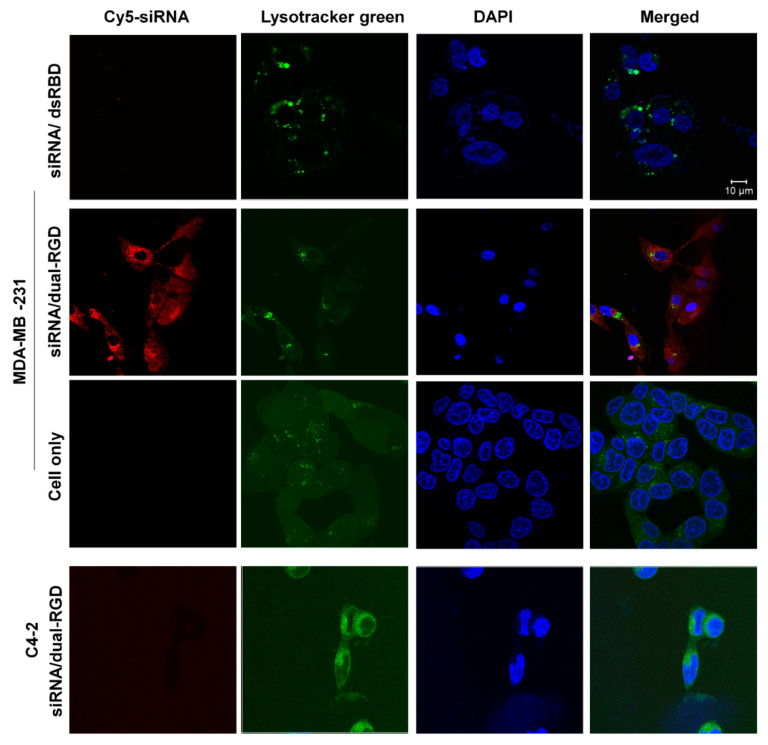
Detection of dual-RGD/Cy5-siRNA internalization by confocal microscopy. The complexes of Cy5-siRNA/dual-RGD were incubated with MDA-MB-231 (integrin α_v_β_3_ positive) and C4-2 cells (integrin α_v_β_3_ negative), and Cy5-siRNA complexed with dsRBD-18H without RGD as a protein control were incubated with MDA-MB-231 cells. LysoTracker Green (green) was used to show endosomes and lysosomes, and DAPI (blue) was used to display the nucleus. Confocal laser scanning microscopy images were acquired after treatment. Scale bar, 10 µm.

**Figure 5 pharmaceutics-13-02182-f005:**
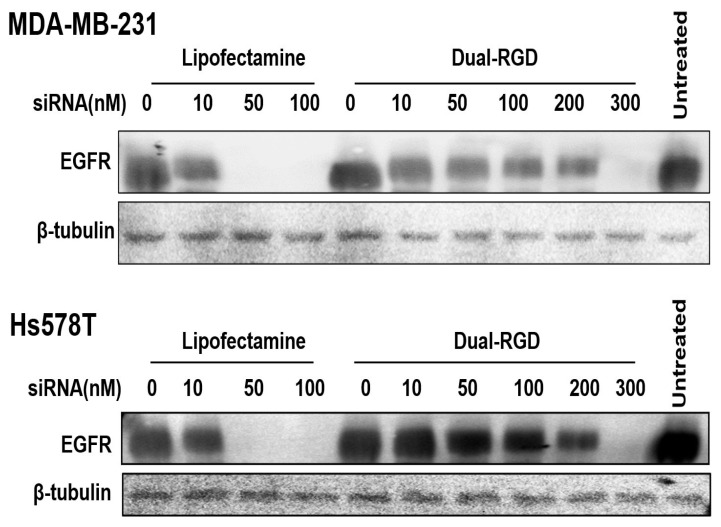
Detection of in vitro target gene knockdown. Integrin α_v_β_3_-expressing MDA-MB-231 and Hs578 T cell lines were treated with EGFR siRNA/dual-RGD at varying concentrations for 72 h. Lipofectamine was selected as a control. After treatment, the cell lysates were probed by Western blot.

**Figure 6 pharmaceutics-13-02182-f006:**
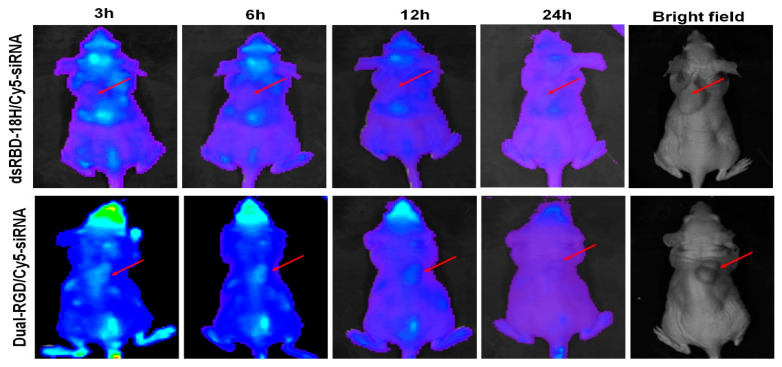
Time-course imaging of Cy5-siRNA/dual-RGD in vivo biodistribution. MDA-MB-231 tumor-bearing mice were iv-treated with complexes of Cy5-siRNA/dual-RGD or Cy5-siRNA/dsRBD-18His. Biodistribution was captured with Xenogen IVIS 100 system.

**Figure 7 pharmaceutics-13-02182-f007:**
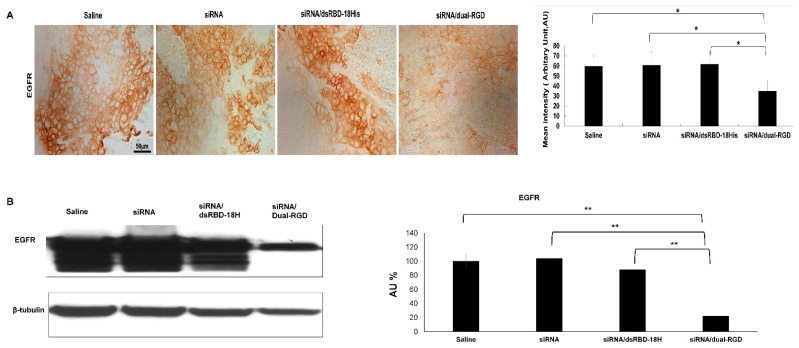
Evaluation of target gene knockdown in vivo. MDA-MB-231 cell xenografts were treated with EGFR siRNA/dual-RGD or controls twice a week for four weeks. (**A**) EGFR expression was detected by IHC. Formalin-fixed, paraffin-embedded sections of xenograft tumors were stained with anti-EGFR antibodies compared with controls of saline, siRNA, and siRNA/dsRBD-18His. IHC measurement was performed with ImageJ plugin IHC profiler. Scale bar, 50 µm. * *p* < 0.05 (**B**) Tumor EGFR protein expression was measured by Western blot. Quantification of protein levels normalized by GAPDH using ImageJ. The results are the pool of three mice per group. ** *p* < 0.01.

**Figure 8 pharmaceutics-13-02182-f008:**
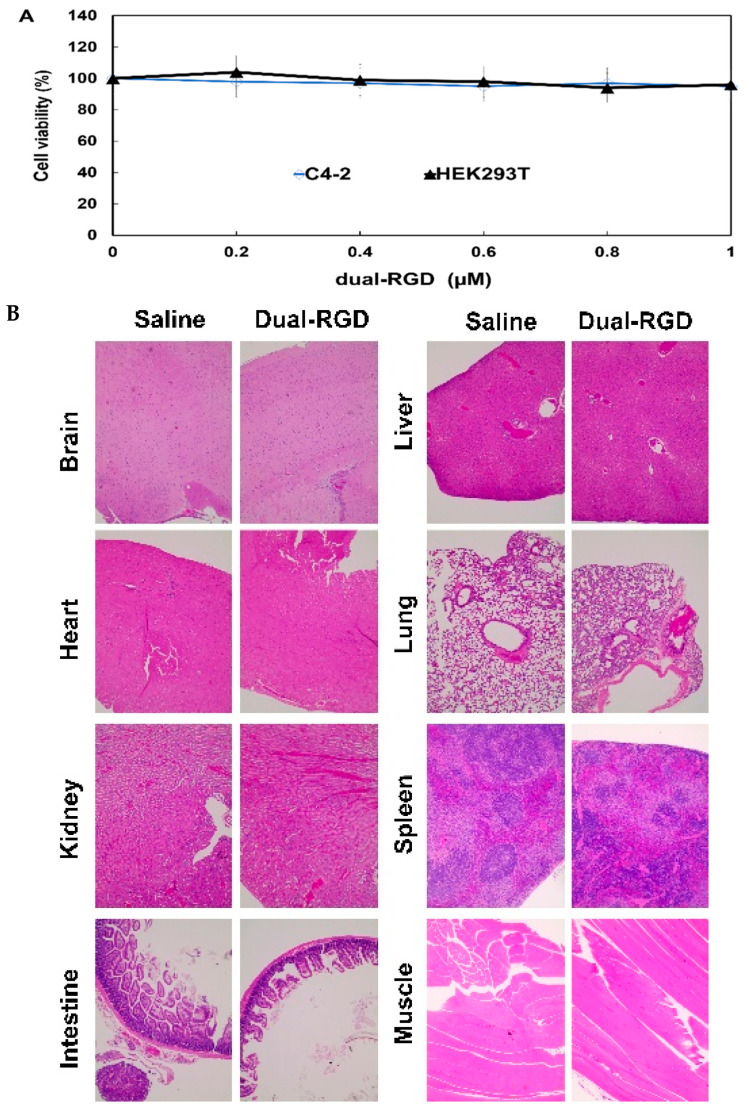
Assessment of toxicity. (**A**) Cell viability assay. Integrin α_v_β_3_-negative C4-2 and HEK293 cells were treated with the varying concentrations of dual-RGD for 72 h. CCK-8 reagent was added into each well for 4 h. Absorbance was measured at 450 nm. (**B**) Histology examination of dual-RGD on major normal organs. Athymic mice were treated with dual-RGD or saline for four weeks. After treatment, the major organs were fixed and stained with H&E. Images were captured under 20× magnification microscopy.

## Data Availability

Not applicable.

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
