# Peer review of "Non-Cationic RGD-Containing Protein Nanocarrier for Tumor-Targeted siRNA Delivery"

_pharmaceutics, 2021, doi:10.3390/pharmaceutics13122182_

Round 1

Reviewer 1 Report

The authors studied and characterized about delivery carrier using RNA binding protein based on non-cationic polymer for siRNA delivery. In addition, in order to produce a more improved carrier, improved endosome escape ability (18His tag)and delivery to selective target cells (RGD) were induced to give increased delivery capability and selectivity in tumors. Based on the results from previous studies, it is a good study to develop an improved siRNA carrier and link it to tumor-specific treatment. The characteristics of the developed carrier and tumor selective results are well performed, but some issues need to be addressed in order for the manuscript to be published in Pharmaceutics. The reviewer offers following comments for improving the manuscript.

⦿ Major comments:

1. The authors developed RNA carriers using protein bonds, not charge bonds, for siRNA delivery in 3.1 section of Results part. However, dual-RGD-dsRBD-18H has a negative charge by the zetaSizer. The siRNA bound here also has a negative charge. Eventually, the formed complex has a negative charge as a whole, and isn't this characteristic likely to cause RGD to reduce the transfer efficiency into the target cell? Therefore, it would exclude selectivity and reduce cellular uptake efficiency than lipofectamine.

2. In MDA-MB-231cell result of Figure 4, the non-RGD conjugates (siRNA/dsRBD-18H) does not enter into the cell well. Nevertheless, green fluorescence is well observed by lysotracker, an endosome staining marker. On the other hand, siRNA/dual-RGD complex has low fluorescence staining of the lysotracker while siRNA is well observed in cells. Why is the lysotracker fluorescence expression low in siRNA/dual-RGD treated cells compared with that of siRNA/dsRBD-18H group ? Both groups have 18H for improved endosome escape.

In addition, the lysotracker fluorescence of siRNA/dual-RGD in C42 cells is well seen. To discuss these results, it is necessary to show the same results of the non-treated cell (cell only) group.

3. In the results of Figure 6, you carried out the biodistribution experiment to evaluate tumor-targeting ability of siRNA/dual-RGD. siRNA-RGD injected into mice was observed in some tumors compared with that of control conjugate, but appear to be mainly accumulated in the brain. It is necessary to show the image of the NC mouse to distinguish whether it is actually accumulated in the brain or a natural auto-fluorescence.

4. Authors examined whether or not dual-RGD carrier protein induces toxicity in vivo in Figure 8. However, shouldn't the actual toxicity assessment be performed as a complex in which siRNA is bound and in mice with tumors? Isn't a simple carrier injection naturally resulting in different results depending on the dosage administered? Then, how did the ds-RGD protein concentration in the previous animal experiment be selected ? ex) biodistribution experiment

5. Finally, the final purpose of the authors' study is to evaluate whether the developed siRNA carrier actually induces anti-tumor effects in vivo. Through this, the value of the developed carrier as a therapeutic substance is proven, and it is very regrettable that there is no result of anti-tumor effect. In the Discussion, authors mentioned that this study is next plan, so I hope you carry it out.

⦿ Minor comments:

1. Add the agency approval protocol number for animal testing in 2.2 section of Materials and Methods part.

2. Please write a coherent expression and spelling of words in Manuscript. It is necessary to check the errors in the whole paper. Ex) 1h or 1 h or 24 hours, italics for enzyme names (line 102 on page 3), modifying CO2 to CO2 and the first letter of the sentence is written in English, not Arabic.

3. Please modify Figure 8A in line 426 on page 13 to Figure 8B.

Reviewer 2 Report

The article “Non-Cationic RGD-Containing Protein Nanocarrier for Tumor-Targeted siRNA De-Livery” by Yu et al. describes the preparation and characterization of a tumor-targeted siRNA vector by fusing RGD peptide into dsRBD-18His protein. It is a well-designed experiment. However, there are several issues that need to be fixed before its acceptance in Pharmaceutics.

My primary concerns are:

  • Figure 6: Biodistribution of Cy5-siRNA/dual-RGD was performed using IVIS imaging. However, this is not a confirmatory method. The authors should report comparative quantitative fluorescence data.
  • Figure 7: Please add quantitative data for Figure A. Also, add statistical values to Figure B.
  • Lines 496-497: The statement of “However, cationic lipofectamine cannot be used in vivo due to nonspecific binding and cytotoxicity” is not entirely true. There are commercially available cationic reagents for in vivo applications (e.g., Invivofectamine 3.0 Reagent).
  • Report the size and charge data for dual-RGD/siRNA.

Round 2

Reviewer 1 Report

This paper was revised to reflect the comments of the reviewers.

The authors addressed my main concerns and questions.

This paper can be accepted for publication as it is now.

Reviewer 2 Report

The authors have adequately addressed my concerns, and it could be accepted for publication in Pharmaceutics.